# Profiling Delirium Progression in Elderly Patients via Continuous-Time Markov Multi-State Transition Models

**DOI:** 10.3390/jpm11060445

**Published:** 2021-05-21

**Authors:** Honoria Ocagli, Danila Azzolina, Rozita Soltanmohammadi, Roqaye Aliyari, Daniele Bottigliengo, Aslihan Senturk Acar, Lucia Stivanello, Mario Degan, Ileana Baldi, Giulia Lorenzoni, Dario Gregori

**Affiliations:** 1Unit of Biostatistics, Epidemiology and Public Health, Department of Cardiac, Thoracic, Vascular Sciences, and Public Health, University of Padova, 35122 Padova, Italy; honoria.ocagli@unipd.it (H.O.); danila.azzolina@uniuopo.it (D.A.); rrozitasoltanmohamadi@gmail.com (R.S.); roqayeh.aliyariamirabadi@unipd.it (R.A.); daniele.bottigliengo@studenti.unipd.it (D.B.); ileana.baldi@unipd.it (I.B.); giulia.lorenzoni@unipd.it (G.L.); 2Department of Translational Medicine, University of Piemonte Orientale, 13100 Vercelli, Italy; 3Department of Actuarial Sciences, Hacettepe University, Ankara 06800, Turkey; aslihans@hacettepe.edu.tr; 4Health professional Management Service (DPS) of the University Hospital of Padova, 35128 Padova, Italy; lucia.stivanello@aopd.veneto.it (L.S.); mario.degan@aopd.veneto.it (M.D.)

**Keywords:** Cox model, continuous-time Markov multi-state transition model, 4AT scale, delirium

## Abstract

Poor recognition of delirium among hospitalized elderlies is a typical challenge for health care professionals. Considering methodological insufficiency for assessing time-varying diseases, a continuous-time Markov multi-state transition model (CTMMTM) was used to investigate delirium evolution in elderly patients. This is a longitudinal observational study performed in September 2016 in an Italian hospital. Change of delirium states was modeled according to the 4AT score. A Cox model (CM) and a CTMMTM were used for identifying factors affecting delirium onset both with a two-state and three-state model. In this study, 78 patients were enrolled and evaluated for 5 days. Both the CM and the CTMMTM show that urine catheter (UC), aging, drugs, and invasive devices (ID) are risk factors for delirium onset. The CTMMTM model shows that transition from no-delirium/cognitive impairment to delirium was associated with aging (HR = 1.14; 95%CI, 1.05, 1.23) and neuroleptics (HR = 4.3; 1.57, 11.77), dopaminergic drugs (HR = 3.89; 1.2, 12.6), UC (HR = 2.92; 1.09, 7.79) and ID (HR = 1.67; 103, 2.71). These results are confirmed by the multivariable model. Aging, ID, antibiotics, drugs affecting the central nervous system, and absence of moving ability are identified as the significant predictors of delirium. Additionally, it seems that modeling with CTMMTM may show associations that are not directly detectable with the traditional CM.

## 1. Introduction

The elderly population is increasing [1] and during hospitalization, they often experienced delirium due to their age and disease severity [2,3]. However, two-thirds of cases with delirium are under-recognized [4]. Delirium has a prevalence of 1–2% in the community, 6–56% in general hospital admission [5], and 29–64% in the elderlies admitted in an acute care setting [6]. Delirium onset requires longer hospitalization, extended health care services with increased risks of mortality and morbidity [2,7] in older patients. 

Delirium, as an acute neuropsychiatric syndrome, is characterized by symptoms fluctuation, shift attention, impaired consciousness, and cognition disturbance like disorientation, memory impairment, and language alteration [5,8]. This mental syndrome may also include disorganized thinking, disturbance in the sleep–wake cycle, and psychomotor activity [8,9]. A complex interaction among predisposing factors (aging, baseline dementia, and functional disabilities) and precipitating ones (medications, surgery, and infection) is responsible for this multifactorial disease [10]. Several hypotheses about delirium pathophysiology have been reported. However, the underlying mechanism is still unclear [11]. The identification and modification of precipitating factors may help in reducing delirium onset [3,12]. Recently, substantial attention has been focused on drugs and devices frequently used in the hospital setting that may induce delirium [10,12,13,14,15].

In literature, delirium outcome has been mainly addressed in prevalence [15,16,17] and incidence studies [18]. These approaches are useful to investigate the magnitude of the problem in a specific ward and moment. Delirium is also considered as a risk factor for clinical outcomes such as mortality in several categories of subjects; for example, COVID-19 patients [19], vascular surgery patients [20], oncological [21], and critically ill adults [22]. Delirium has also been considered a factor influencing the risk of developing other new forms of dementia [23] and as a possible issue affecting the quality of life [24]. In these research studies, the statistical methods mostly used are linear regression [16] or Cox proportional hazard model (shortly Cox model) [16,21,22]. Such methods are relevant for understanding the problem of delirium and its consequences in terms of clinical outcomes. For predictive models, instead, logistic regression [25,26] is mostly used, recently with machine learning technique [27]. The Cox model (CM) is used when considering time to define the probability of developing an event. For example, Lee et al. [28] used this model for detection of the role of frailty for delirium onset.

A Cox model analyzes the effects of several variables on survival time [29,30], considered as an event that occurs as a consequence of a previous event. However, in longitudinal studies, the same measurements are often collected in different moments and patients may experience more than two transitions.

A multi-state model (MSM), compared to a Cox model, can parameterize risk through intermediate states throughout the follow-up time. Indeed, a patient may develop intermediate states, passing from the absence of delirium to more or less severe forms of delirium. The MSM model allows to capture the risk that characterizes these transitions considering also the time elapsed from the previous event occurrence [31].

One of the most popular methods in survival analysis is the time-continuous Markov multi-state transition model in which one state depends on the previous one. Hence, MSM may help to define the transition intensities with the hazards for passing from one state to another. This model permits the consideration of covariates through the transition intensities and this helps to explain the differences of the individual in the various states and the effects of single covariate varying in the different states [31]. So, the MSM could help to discover hidden aspects behind the delirium state transition that the Cox model is unable to evaluate.

## 2. Materials and Methods

### 2.1. Aim

This work aims to identify the contributing factors for delirium onset, worsening, and transitions comparing two different statistical methods.

### 2.2. Design

This is a longitudinal observational study performed in September 2016 in three wards of the University Hospital of Padova: orthopedics, geriatrics, and general medicine. The current study is part of a program to create a standardized flow chart for better detection and treatment of delirium in elderly patients in the whole hospital.

### 2.3. Participants

Patients older than 65 years admitted to the wards involved in the study were enrolled. A hospitalization of at least five days and a good understanding of the Italian language were other additional criteria for inclusion in the study. Patients with a psychiatric illness already diagnosed during admission, with any communication problems (such as aphasia, coma status), or with a terminal disease were excluded.

### 2.4. Data Collection

After a maximum of 24 h since admission, a nurse evaluated the risk for delirium in patients considered eligible until the fifth day of hospitalization with the 4AT scale.

### 2.5. Ethical Aspects

Study design and data collection were conducted in full respect of clinical practice regulation. Data were collected in everyday clinical practice, so no further informed consent was needed. For this reason, ethical committee approval was waived.

### 2.6. Instrument

The 4AT scale was used for delirium detection in this study. The instrument is a simple screening tool with high sensitivity and acceptable specificity and requires no specific training [9,32]. This instrument has been used among hospitalized elderly patients, especially the ones with acute medical illnesses. The 4AT scale is based on direct observation of the patient and collection of information from various sources [33,34]. It has 0 to 12 scores: 0 suggests for no delirium, 1–3 is suggestive for cognitive impairment, and ultimately, a score equal to 4 or above suggests delirium [32,33].

### 2.7. Data Analysis

#### 2.7.1. Descriptive Statistics

Descriptive statistics are summarized as follows: If the variable is the continuous median, I and III quartiles are used, and if it is categorical relative and absolute, frequencies are reported. Kruskal–Wallis type tests were performed for continuous variables and the Pearson chi-square test for categorical.

#### 2.7.2. Sample Size

A simulation experiment was carried out for the sample size evaluation. Databases of sample size 79 were generated 400 times by a Cox model assuming an HR of 1.1 and a hospital stay time of 5 days. The data-generating model included a summarized confounding effect in two covariates (beta) including assuming the same HR of 1.1. The Cox model was calculated on all generated databases and the main effect was significant in 80.5% of the simulations.

#### 2.7.3. Variables Collected

For each patient, socio-demographic characteristics at baseline (age, diagnosis, comorbidities, visual and hearing impairment); physical function, physical restraint, presence/absence of invasive device, and drugs of each day were collected. For the analysis, variables were grouped as follows: (i) invasive devices, number of invasive devices (urine catheter (UC), central venous catheter (CVC), peripheral venous catheter (PVC), nasogastric feeding tubes (NG), percutaneous endoscopic gastrostomy (PEG), other device and physical restraint); (ii) basic needs, categorical variables (yes/no) which is “yes” if at least one intervention was made on the basic needs (fever or pain) during the day; (iii) number of drugs affecting the central nervous system (DACNS) (i.e., anticholinergics, dopaminergic, steroids, opioids, antiepileptics, anxiolytics, neuroleptics, and antidepressants); (iv) antibiotics, categorical variables (yes/no), which is equal to “yes” if at least one antibiotic has been administered per day (i.e., quinolones, voriconazole antifungals, and cephalosporin); (v) psychiatric pathology, categorical variables (yes/no), which is equal to “yes” if the patient is affected by dementia or depression or other psychiatric pathology; (vi) mobility aids: number of asking for aids from nursing staff per day, including the chair–bed transfer, walking, or going up/downstairs.

#### 2.7.4. Delirium Modeling

Transition frequencies and probabilities were reported as three states (no delirium, cognitive impairment, and delirium) according to the 4AT score. A model with two states was also created: no delirium (4AT score smaller and equal than 3) and delirium (4AT score higher than four).

Two estimation approaches were used for evaluating the hazard of the transition from one state to another: A CM and a continuous-time Markov multi-state transition model (CTMMTM). The first approach evaluates the effect of the covariates on the transition hazards. The proportionality of hazard was evaluated using proportional hazard tests and diagnostics based on weighted residuals. The second model describes the process in which a patient moves through a series of delirium states in continuous time for longitudinal data. Data consist of observations of the process at arbitrary times so that the exact times when the delirium state changes are unobserved. The next state to which a patient moves and the time of the modification is governed by a set of transition intensities for each pair of delirium states, i and j. A transition matrix has been defined whose rows sum is zero so that the diagonal entries are defined by all transitions between delirium states are permitted (Figure 1).

Statistical analyses were performed using R 3.3.5 [35] with rms [36], survival [37], and msm packages [38].

## 3. Results

In this study, 78 patients were enrolled in general medicine, geriatrics, and orthopedics hospital wards, during the 5-day study period. Table 1 reports the baseline characteristics of our sample population according to 4AT classification.

At admission time, 17 patients experienced delirium, 31 patients reported no delirium, and 30 cognitive impairment. Patients that experienced delirium were older (89) compared with the other two groups (median age of 83 and 86.5), prevalently female (52, 67%) with a lower middle school. Drugs mostly used were steroids and anti-inflammatory (48, 62%) along with drugs for inducing sleeping (46, 64%). Peripheric venous catheters were the device mostly used in these patients (76, 97%) along with physical restraints (58, 74%). Dementia history was the comorbidities mostly present in all the groups. Hearing and visual impairments, respectively (10, 59%) and (9, 53%), were mainly present in patients at risk for delirium.

### 3.1. Transition Frequencies and Probabilities

The transition frequencies and probabilities for three-state and two-state delirium are reported in Table 2.

The highest number of transitions in the three-state case were observed from cognitive impairment to delirium with 18 episodes and its opposite direction with 13 individuals with a transition probability of 17% in both directions. In two-state cases instead, 19 episodes were observed to pass from no-delirium/cognitive impairment to delirium and 15 in the opposite direction. In this case, the probability of passing in the first direction is very low 8% (95% CI: 0.05, 0.12) and in the opposite of 20% (95% CI: 0.13, 0.31).

### 3.2. Cox model and Continuous-Time Markovian Multi-State Transition Model

The results of the two state-delirium models both for the Cox model and the continuous-time Markovian multi-state transition model were compared and reported in Table 3.

In both models, the urine catheter increased the transition from no-delirium/cognitive impairment to delirium (HR = 20.38; 95% CI: 2.72, 152.9) in the Cox model and 2.92 (95% CI: 1.09, 7.79) in the MSM model. Additionally, age and a higher number of invasive devices increase the transition from no-delirium/cognitive impairment to delirium in both models, similar with strength. Drugs such as neuroleptic (HR = 4.3; 95% CI, 1.57, 11.77) and dopaminergic (HR = 3.89; 95% CI, 1.2, 12.6) increased the hazard from no-delirium/cognitive impairment to delirium in the MSM, in the Cox model, drugs influence the increase of the hazard when grouped in DACNS.

### 3.3. Multivariable Analysis

Multivariable analysis of the Cox model (Table 4) showed that older individuals (HR = 1.11; 95% CI, 1.03, 1.2) and subjects with more use of invasive devices (HR = 1.83; 95% CI, 1.13, 2.97) had a higher hazard risk of moving from no-delirium/cognitive impairment to delirium state.

According to our multivariable Markovian findings, delirium risk among subjects with no-delirium/cognitive impairment increased with increasing of age (HR = 1.12; 95% CI, 1.03, 1.21).

## 4. Discussion

Although delirium frequently occurs among the hospitalized elderly population, its understanding is still insufficient among caregiver providers.

Our results are in agreement with the literature findings concerning the associations between delirium and the use of invasive devices, [14,15] utilization of drugs affecting the central nervous system such as neuroleptic and anti-depressive [3,15,30,39,40], administration of antibiotics [10], and age [2,10,11,41,42]. For what concerns drugs, delirium seems to be caused by brain activities in dopaminergic overflow and/or anticholinergic deficiency [10] and psychoactive drugs work in the same neurotransmitter pathways. Another relevant factor influencing delirium onset is aging, which produces physiological changes such as cholinergic system atrophy [43]. Moreover, elderly people face multi-morbidity conditions and consequently, the need for multiple drug therapy will increase [44]. Hence, synergetic drug reactions, their relevant adverse events may cause aging-induced pharmacokinetic alteration to effect treatment or even exacerbate the condition [44]. These results are confirmed by both models and the multivariable analysis, with some differences for what concerns the strength of the HR.

The level of mobilization is also a factor that influences the delirium onset, our results show that patients who had a higher level of functional activity (i.e., walking), had a lower probability of facing delirium as already found by in Solà-Miravete [45].

Our results also show that there are numerous transitions between cognitive impairment and delirium states. This phenomenon is remarkable, considering the short follow-up period. This could mean that in longer period, patients may experience different delirium status depending on different moments, for example, after a procedure or administration of a new medication. This multiplicity of delirium status in a single patient should be taken into consideration when working on predictive models for delirium detection or for studies that consider risk factors for delirium. Cognitive impairment in elderlies increases the probability of several acute medical problems and might lead to poor prognosis, and as reported by a systematic review, it is a strong risk factor for delirium onset [11]. It has been reported that 20% of delirium prevalence was among older patients with cognitive impairment [46]. So, early detection following with pharmacological and non-pharmacological approaches can prevent it [47].

The studies on delirium are usually based on a single-point assessment, and they usually do not include the changes that patients underwent during their hospital stay. A recent study has developed a “dynamic” predictive tool for delirium in ICU patients [26]. In this study, the authors consider all the risk factors for delirium that “could come and go prior to the onset of delirium”. However, even in this case, delirium evaluation was based on a single assessment. Hence, the use of models that consider time in their structure could be helpful to better understand delirium progression, as our work suggests.

Both methods are suitable in the condition of delirium evolution, both show similar results in identifying risk factors for delirium onset. However, some differences have to be taken into account when comparing these two methods. In the Cox model, which is appropriate for survival analysis with censored data and time-varying covariates, the outcomes depend only on prognostic factors [48]. It also considers the concepts of interaction and collinearity of independent variables [49]. MSM provides a broader biological understanding about obscure aspects of disease endpoints, which are commonly away from caregivers’ attention during the study [50]. As in our case, the Markovian multi-state model is helpful in situations where a patient can experience different states and can make transitions between the states. This model is a more realistic tool for comparison with the discrete-time model since it would allow transitions among states to happen at any time of the follow-up. Moreover, since it is a continuous-time model, it is preferable as it takes into account the transitions that may have a small probability [51]. Thus, in delirium progression, the continuous-time Markov multi-state transition model as a flexible approach supported the Cox model, which is more naïve but essential in addressing the needs of health care professionals. Ultimately, the utilization of 4AT with good sensitivity and acceptable specificity has simplified delirium detection due to no complicated training needs [40].

### Limitations

These results should be interpreted carefully, due to a couple of limitations. To begin with, the small sample size of this study should be taken into consideration with caution, as it does not allow to use more collected risk factors in our models. The small follow-up period of observation might also be an obstacle for the identification of delirium which can be experienced with high frequency in a longer follow-up.

## 5. Conclusions

According to our results, older age, drugs, particularly those affecting the central nervous system, and invasive devices play an important role in delirium onset. Except for age, drugs and invasive devices are modifiable risk factors, and they should be carefully prescribed and used in this frail population. Moreover, by obtaining the pharmaceutical history of patients in the admission time and identification of any drugs as the potential reasons for delirium or whether it can worsen its condition, the administration should be stopped and an alternative pharmaceutical one should be prescribed.

In conclusion, in this study, we adopted two different statistical approaches to model the change in delirium status of a group of elderly patients admitted to hospital experiencing different stages of the 4AT scale. The choice of considering several states and related transitions can provide a set of more detailed information if compared with an approach that considers a single endpoint, such as the Cox model, where it is not possible to include an intermediate state such as the cognitive impairment, which is a frequently observed state and also a risk factor for delirium onset. In this sense, MSM shows associations that are not directly detectable with the Cox model. MSM model may be helpful with a larger cohort of patients for the prediction of incidence and prevalence over the larger time horizon.

## Figures and Tables

**Figure 1 jpm-11-00445-f001:**
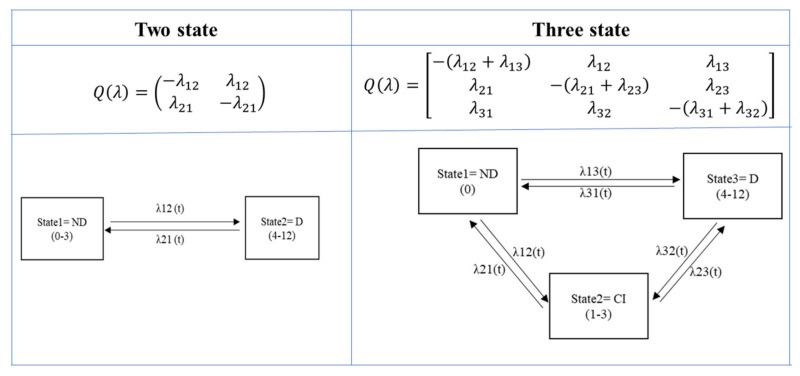
The Q transition intensity matrix for two states and three states modeling. The off diagonal elements in Q are rates at which subjects move into other delirium states, while the diagonal elements are transition probabilities at which subjects remain in their state. Transitions between all the states are possible.

**Table 1 jpm-11-00445-t001:** Patients’ characteristics according to baseline delirium. Continuous data were reported as median (I, III quartiles), categorical data were reported as absolute number (percentage). Kruskal–Wallis type tests were performed for continuous variables and the Pearson chi-square test for categorical.

Variables		No Delirium	Cognitive Impairment	Delirium	Total	*p*-Value
		(*n* = 31)	(*n* = 30)	(*n* = 17)	(*n* = 78)	
Female gender		19 (61%)	22 (73%)	11 (65%)	52 (67%)	0.597
Age (year)		83 (76.0, 86.50)	86.5 (82.25, 90.0)	89 (83.0, 94.0)	85 (80.0, 89.0)	0.004 *
Age (class)	<70	4 (13%)	2 (7%)	0 (0%)	6 (8%)	0.076
	71–75	3 (10%)	0 (0%)	2 (12%)	5 (6%)	
	76–80	6 (19%)	4 (13%)	1 (6%)	11 (14%)	
	81–85	8 (26%)	8 (27%)	3 (18%)	19 (24%)	
	86–90	8 (26%)	10 (33%)	3 (18%)	21 (27%)	
	91–95	1 (3%)	6 (20%)	6 (35%)	13 (17%)	
	>95	1 (3%)	0 (0%)	2 (12%)	3 (4%)	
Education	Up to school degree	27 (87%)	28 (93%)	14 (82%)	69 (88%)	0.503
	University degree	4 (13%)	2 (7%)	3 (18%)	9 (12%)	
Hospital wards	General medicine	16 (52%)	5 (17%)	3 (18%)	24 (31%)	<0.001 *
	Geriatrics	1 (3%)	17 (57%)	6 (35%)	24 (31%)	
	Orthopedics	14 (45%)	8 (27%)	8 (47%)	30 (38%)	
Drugs	Anticholinergics	4 (13%)	6 (20%)	2 (12%)	12 (15%)	0.667
	Dopaminergic	1 (3%)	4 (13%)	0 (0%)	5 (6%)	0.13
	Steroids and anti-inflammatory	18 (58%)	20 (67%)	10 (59%)	48 (62%)	0.762
	Opioids	8 (26%)	8 (27%)	8 (47%)	24 (31%)	0.258
	Antianxiety and benzodiazepine	11 (35%)	10 (33%)	6 (35%)	27 (35%)	0.982
	Neuroleptics	1 (3%)	5 (17%)	8 (47%)	14 (18%)	<0.001 *
	Anti-depressives	3 (10%)	7 (23%)	6 (35%)	16 (21%)	0.097
	Deprivation sleep	17 (63%)	16 (57%)	13 (76%)	46 (64%)	0.421
	Cephalosporin antibiotics	2 (6%)	7 (23%)	0 (0%)	9 (12%)	0.029 *
	Quinolone antibiotics	4 (13%)	4 (13%)	0 (0%)	8 (10%)	0.288
	Antiepileptic levetiracetam	1 (3%)	1 (3%)	2 (12%)	4 (5%)	0.374
Devices	Restraints	12 (39%)	29 (97%)	17 (100%)	58 (74%)	<0.001 *
	UC	9 (29%)	16 (53%)	14 (82%)	39 (50%)	0.002
	CVC	0 (0%)	2 (7%)	2 (12%)	4 (5%)	0.186
	PVC	31 (100%)	28 (93%)	17 (100%)	76 (97%)	0.194
	NG	0 (0%)	2 (7%)	1 (6%)	3 (4%)	0.354
	PEG	0 (0%)	1 (3%)	0 (0%)	1 (1%)	0.445
	Other devices (N)	1 (3%)	3 (10%)	4 (24%)	8 (10%)	0.085
Other devices	Colostomy	30 (97%)	28 (93%)	13 (76%)	71(91%)	0.139
	Drainage elastomer	0 (0%)	1 (3%)	3 (18%)	4 (5%)	
	Physical restraint	0 (0%)	0 (0%)	1 (6%)	1 (1%)	
	Vac therapy	1 (3%)	0 (0%)	0 (0%)	1 (1%)	
	Valve	0 (0%)	1 (3%)	0 (0%)	1 (1%)	
Comorbidities	Dementia	2 (6%)	8 (27%)	8 (47%)	18 (23%)	0.005 *
	Alcoholism	1 (3%)	0 (0%)	0 (0%)	1 (1%)	0.464
	Drugs addiction	1(3%)	0 (0%)	0 (0%)	1 (1%)	0.464
	Depression	2 (6%)	4 (13%)	0 (0%)	6 (8%)	0.243
	Previous delirium	0 (0%)	2 (7%)	1 (6%)	3 (4%)	0.354
	Other psychiatry pathologies	1 (3%)	0% (0)	2 (12%)	3 (4%)	0.128
	Diabetes	7 (23%)	4 (13%)	5 (29%)	16 (21%)	0.395
	Cancers	10 (32%)	11 (37%)	8 (47%)	29 (37%)	0.596
	Malnutrition-Dehydration	10 (32%)	13 (43%)	7 (41%)	30 (38%)	0.651
	Surgery history	0 (0%)	1 (3%)	3 (18%)	4 (5%)	0.025
	Previous admission	27 (87%)	29 (97%)	16 (94%)	72 (92%)	0.356
	Visual disabilities	13 (42%)	7 (23%)	9 (53%)	29 (37%)	0.102
	Hearing disabilities	8 (26%)	11 (37%)	10 (59%)	29 (37%)	0.077
Bed to chair transferring ability	0	13 (42%)	20 (67%)	16 (94%)	49 (63%)	<0.001 *
	5	7 (23%)	9 (30%)	1 (6%)	17 (22%)	
	10	11 (35%)	1 (3%)	0 (0%)	12 (15%)	
Walking ability	0	13 (42%)	24 (80%)	17 (100%)	54 (69%)	<0.001 *
	5	9 (29%)	5 (17%)	0 (0%)	14 (18%)	
	10	9 (29%)	1 (3%)	0 (0%)	10 (13%)	
Stairs going down ability	0	21 (68%)	28 (93%)	17 (100%)	66 (85%)	0.02 *
	5	5 (16%)	1 (3%)	0 (0%)	6 (8%)	
	10	5 (16%)	1 (3%)	0 (0%)	6 (8%)	
Pain		19 (61%)	14 (47%)	7 (41%)	40 (51%)	0.334
Fever		6 (19%)	5 (17%)	5 (29%)	16 (21%)	0.57
ICD diagnosis	Circulatory	16 (52%)	20 (67%)	8 (47%)	44 (56%)	0.2
	Musculoskeletal and connective tissue	14 (45%)	8 (27%)	8 (47%)	30 (38%)	
	Digestive	0 (0%)	0(0%)	1(6%)	1 (1%)	
	Respiratory	1 (3%)	0 (0%)	0 (0%)	1 (1%)	
	Symptoms, signs, and undefined morbidity states	0 (0%)	2 (7%)	0 (0%)	2 (3%)	

* *p*-value < 0.001; Abbreviation; CI: cognitive impairment, UC: urine catheter, CVC: central venous catheter, PVC: peripheral venous catheter, NG: nasogastric tube, PEG: percutaneous endoscopic gastrostomy, DACNS: drugs affecting central nervous system.

**Table 2 jpm-11-00445-t002:** The observed number of transitions and transitions probabilities (%; 95% CI) for three and two delirium states.

Three Delirium States				Two Delirium States		
	No Delirium	Cognitive Impairment	Delirium		No Delirium—Cognitive Impairment	Delirium
**No delirium**	124 (95; 0.40–0.97)	6 (5; 2–11)	1 (1; 0.3–0.52)	**No delirium—Cognitive impairment**	218 (92; 88–95)	19 (8; 5–12)
**Cognitive impairment**	8 (8; 3–14)	80 (75; 65–83)	18 (17; 11–26)	**Delirium**	15 (20; 13–31)	60 (80; 70–87)
**Delirium**	2 (3; 1–13)	13 (17; 11–28)	60 (80; 66–87)			

**Table 3 jpm-11-00445-t003:** Proportional transition Cox hazards and continuous-time Markovian multi-state transition two-states model (95% CI). The proportionality of hazard has been evaluated using proportional hazards tests and diagnostics based on weighted residuals.

	Proportional Transition Cox Hazard Model		Continuous-Time Markovian Multi-State Transition Model	
	ND-CI to D	D to ND-CI	ND-CI to D	D to ND-CI
	HR (95%CI)	HR (95%CI)	HR (95%CI)	HR (95%CI)
Urine catheter	20.38 (2.72, 152.9)	2.2 (0.49, 9.81)	2.92 (1.09, 7.79)	1.8 (0.4, 8.05)
Bed-Chair transfer	0.76 (0.61, 0.94)	1.09 (0.81, 1.48)	0.86 (0.73, 1)	0.91 (0.6, 1.38)
Walking	0.82 (0.68, 0.99)	1.36 (0.88, 2.11)	0.86 (0.72, 1.03)	1.05 (0.7, 1.57)
Sitting and standing scale	0.88 (0.72, 1.07)	1.61 (1.01, 2.55)	0.9 (0.64, 1.25)	1.28 (0.87, 1.89)
Other devices number	1.24 (0.41, 3.79)	0.93 (0.32, 2.73)	1.61 (0.52, 4.96)	1.18 (0.4, 3.52)
Pain	1.25 (0.5, 3.09)	0.7 (0.25, 2.01)	1.48 (0.58, 3.73)	0.96 (0.34, 2.68)
Fever	1.16 (0.33, 4.02)	0.19 (0.33, 1.48)	1.35 (0.38, 4.71)	0.88 (0.27, 2.8)
Opioids	1.93 (0.78, 4.77)	1.25 (0.45, 3.45)	2.2 (0.88, 5.53)	1.25 (0.45, 3.52)
Neuroleptics	2.21 (0.73, 6.72)	1.15 (0.4, 3.29)	4.3 (1.57, 11.77)	1.28 (0.44, 3.57)
Antianxiety and Benzodiazepine	1.05 (0.42, 2.68)	0.83 (0.28, 2.43)	0.46 (0.15, 1.39)	0.75 (0.25, 2.21)
Anti-depressives	1.97 (0.71, 5.49)	0.89 (0.28, 2.81)	2.6 (0.96, 7.09)	1.14 (0.38, 3.42)
Anticholinergics	1.72 (0.69, 4.78)	0.92 (0.21, 4.12)	1.33 (0.43, 4.13)	1.02 (0.22, 4.6)
Dopaminergics	2.63 (0.78, 9.07)	0.89 (0.11, 6.9)	3.89 (1.2, 12.6)	1.13 (0.14, 9.28)
Education condition	0.5 (0.07, 3.78)	0.31 (0.04, 2.38)	0.46 (0.06, 3.47)	0.3 (0.04, 2.32)
Dementia	2.34 (0.89, 6.16)	0.89 (0.32, 2.51)	2.37 (0.88, 6.38)	0.93 (0.33, 2.68)
Sex	0.97 (0.37, 2.54)	0.63 (0.2, 2)	0.91 (0.34, 2.42)	0.61 (0.19, 1.94)
Age	1.13 (1.05, 1.22)	0.99 (0.92, 1.06)	1.14 (1.05, 1.23)	1 (0.93, 1.07)
Diabetes	0.79 (0.23, 2.7)	1.09 (0.35, 3.42)	0.8 (0.23, 2.79)	1.07 (0.34, 3.41)
Psychiatric pathologies	1.71 (0.67, 4.34)	1.02 (0.37, 2.81)	1.74 (0.67, 4.49)	1.03 (0.37, 2.89)
Invasive devices (number)	2.06 (1.28, 3.3)	0.81 (0.4, 1.66)	1.67 (1.03, 2.71)	0.79 (0.37, 1.7)
Basic needs	1.21 (0.48, 3.03)	0.47 (0.17, 1.33)	1.45 (0.56, 3.75)	0.95 (0.32, 2.82)
DACNS	1.44 (1.03, 2)	1 (0.64, 1.56)	1.36 (0.94, 1.95)	1.01 (0.65, 1.58)
DACNS (number)	2.24 (0.9, 5.58)	1.24 (0.45, 3.42)	2.46 (0.97, 6.25)	1.19 (0.42, 3.35)
Antibiotics	2.63 (1.05, 6.54)	1.13 (0.32, 4.02)	1.67 (0.62, 4.5)	0.91 (0.2, 4.12)
Mobility aids	0.85 (0.14, 0.99)	0.95 (0.14, 1.31)	0.94 (0.87, 1.02)	1.03 (0.86, 1.23)

Abbreviation; ND: no delirium, CI: cognitive impairment, D: delirium, HR: hazard ratio.

**Table 4 jpm-11-00445-t004:** Multivariable proportional transition Cox hazards model and continuous-time Markovian multi-state transition model (two-states, 95% CI).

	No Delirium-Cognitive Impairment to Delirium	Delirium to No Delirium-Cognitive Impairment
	**HR (95% CI)**	**HR (95% CI)**
**Variables**	**Proportional Transition Cox model**	
Age	1.11 (1.03, 1.2)	0.98 (0.91, 1.06)
Invasive devices (number)	1.83 (1.13, 2.97)	0.78(0.37, 1.62)
	**Continuous-Time Markovian Multi-state**	
Age	1.12 (1.03, 1.21)	0.99 (0.92, 1.07)
Invasive devices (number)	1.43 (0.85, 2.38)	0.78 (0.36, 1.69)

Abbreviation; CI: cognitive impairment.

## Data Availability

The data presented in this study are available on request from the corresponding author. The data are not publicly available due to privacy.

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
