# Peer review of "Profiling Delirium Progression in Elderly Patients via Continuous-Time Markov Multi-State Transition Models"

_jpm, 2021, doi:10.3390/jpm11060445_

Round 1
Reviewer 1 Report
The main point of this study is to suggest a more useful statistical method for delirium research.
Then, prior to presenting the usefulness in delirium research with the two statistical methods in the introduction, I think that the main statistical trends used in delirium research and problems with it should be presented.
In fact, I think that the most problems in this study are two things the authors have suggested in terms of limitations: insufficient subjects and observations over a short period of time.
In addition to presenting it as a limitation, I think that descriptions to offset this limitation should be made in the introduction and discussion.
I believe the Cox model as a survival model is a model that is not generally available in terms of delirium progression and detection, given the types of delirium with varying incidence and severity in the clinical setting. So I think there should be some background explanation so that the reader can understand why the author tried to compare the Cox model and the Markov multi-state transition model in profiling delirium progression.
In the introduction, I think the reason why this study was attempted should be presented more logically. In the discussion, it is necessary to present more concretely how readers can apply and utilize the meaning of this research result in their delirium researches.
The following is minor, and it is recommended to modify the following items in the manuscript description.
1. Table1. Wilcoxon-type tests: It is correct to use a nonparametric test, but the Wilcoxon-test is used to compare two groups. Since this study compared three different groups (No delirium, Cognitive impairment, Delirium), different analysis methods are needed.
2. Table1. In the case of the p-value of, it is necessary to indicate “*” in the significant value.
3. Table2. Observed transition is required to indicate (n) and transition probabilities (%).
4. Table3. 95% CI format change (2.72-152.9) -> (2.72,152.9)
5. Table4. It is necessary to succinctly modify the table.
HR Lower Upper -> HR (95% CI)
6. Page 3 word correction (azard test -> hazard test)
7. Correction of notation on page 7
“(95% CI: 1.09-7.79) in the MSM model. -> (95% CI: 1.09,7.79)
Author Response
First reviewer
Open Review
(x) I would not like to sign my review report
( ) I would like to sign my review report
English language and style
( ) Extensive editing of English language and style required
( ) Moderate English changes required
( ) English language and style are fine/minor spell check required
(x) I don't feel qualified to judge about the English language and style
Yes Can be improved Must be improved Not applicable
Does the introduction provide sufficient background and include all relevant references?
( ) ( ) (x) ( )
Is the research design appropriate?
(x) ( ) ( ) ( )
Are the methods adequately described?
(x) ( ) ( ) ( )
Are the results clearly presented?
( ) (x) ( ) ( )
Are the conclusions supported by the results?
( ) ( ) (x) ( )
Comments and Suggestions for Authors
The main point of this study is to suggest a more useful statistical method for delirium research.
We thank the reviewer for the careful consideration and overall positive judgement given to our work.
Then, prior to presenting the usefulness in delirium research with the two statistical methods in the introduction, I think that the main statistical trends used in delirium research and problems with it should be presented.
Thanks for the suggestion. In literature, the delirium outcome has been mainly addressed in prevalence and incidence studies. Cox model is used when investigating risk factors both for developing delirium and considering delirium as a risk factor for other healthcare outcomes. We have added these information on the introduction (lines 61-74).
In fact, I think that the most problems in this study are two things the authors have suggested in terms of limitations: insufficient subjects and observations over a short period of time.
In addition to presenting it as a limitation, I think that descriptions to offset this limitation should be made in the introduction and discussion.
As suggested by the reviewer this part was improved in the discussion (lines 910-915).
I believe the Cox model as a survival model is a model that is not generally available in terms of delirium progression and detection, given the types of delirium with varying incidence and severity in the clinical setting. So I think there should be some background explanation so that the reader can understand why the author tried to compare the Cox model and the Markov multi-state transition model in profiling delirium progression.
The Cox model is used for detection of risk factors, or when delirium is considered as a risk factor for other outcomes. We have decided to compare Cox model and Markov multi-state transition model since they both considered time from the occurrence of an event to another. (Lines 61-83)
In the introduction, I think the reason why this study was attempted should be presented more logically.
Thanks for pointing this out. Delirium is a phenomenon that could occur more times in a single episode of hospitalization. Therefore, delirium progression models should consider time. Clarified in the introduction (lines 79-83).
In the discussion, it is necessary to present more concretely how readers can apply and utilize the meaning of this research result in their delirium researches.
Delirium is a “dynamic” phenomenon, so the need of models that can incorporate this characteristic (lines 973-989).
The following is minor, and it is recommended to modify the following items in the manuscript description.
- Table1. Wilcoxon-type tests: It is correct to use a nonparametric test, but the Wilcoxon-test is used to compare two groups. Since this study compared three different groups (No delirium, Cognitive impairment, Delirium), different analysis methods are needed.
We agree with the reviewer. There was a mistake in reporting the wording of the test. A Kruskal-Wallis test was performed. The changes have been reported in the manuscript.
- Table1. In the case of the p-value of, it is necessary to indicate “*” in the significant value.
Modified as suggested.
- Table2. Observed transition is required to indicate (n) and transition probabilities (%).
Done.
- Table3. 95% CI format change (2.72-152.9) -> (2.72,152.9)
Done, modified as suggested.
- Table4. It is necessary to succinctly modify the table.
HR Lower Upper -> HR (95% CI)
Thanks, modified as suggested.
- Page 3 word correction (azard test -> hazard test)
Done.
- Correction of notation on page 7
“(95% CI: 1.09-7.79) in the MSM model. -> (95% CI: 1.09,7.79)
Modified as suggested.
Submission Date
25 April 2021
Date of this review
30 Apr 2021 06:37:45
Reviewer 2 Report
This is an interesting manuscript regarding profiling delirium progression in Elderly patients via new models. The manuscript is well written. But as I'm not a statistician,I can't tell if this analysis (Continuous-Time Markov Multistate Transition model) is appropriate for this study. I think statisticians should also be included in the reviewers. In addition, is the sample size large enough to be analyzed with the Cox Hazard model?Author Response
Second reviewer
We thank the reviewer for the careful consideration and the suggestion given to our work.
Open Review
(x) I would not like to sign my review report
( ) I would like to sign my review report
English language and style
( ) Extensive editing of English language and style required
( ) Moderate English changes required
(x) English language and style are fine/minor spell check required
( ) I don't feel qualified to judge about the English language and style
|
Yes |
Can be improved |
Must be improved |
Not applicable |
|
|
Does the introduction provide sufficient background and include all relevant references? |
(x) |
( ) |
( ) |
( ) |
|
Is the research design appropriate? |
( ) |
( ) |
( ) |
(x) |
|
Are the methods adequately described? |
(x) |
( ) |
( ) |
( ) |
|
Are the results clearly presented? |
(x) |
( ) |
( ) |
( ) |
|
Are the conclusions supported by the results? |
(x) |
( ) |
( ) |
( ) |
Comments and Suggestions for Authors
This is an interesting manuscript regarding profiling delirium progression in Elderly patients via new models. The manuscript is well written. But as I'm not a statistician,I can't tell if this analysis (Continuous-Time Markov Multistate Transition model) is appropriate for this study. I think statisticians should also be included in the reviewers. In addition, is the sample size large enough to be analyzed with the Cox Hazard model?
Thanks for pointing this out. A simulation experiment was carried out for the sample size evaluation. Databases of sample size 79 were generated 400 times by a Cox model assuming an HR of 1.1 and a hospital stay time of 20 days. The data-generating model included a summarised confounding effect in two covariates (beta) including assuming the same HR of 1.1. The Cox model was calculated on all generated databases and the main effect was significant in 80.5 % of the simulations.
Submission Date
25 April 2021
Date of this review
07 May 2021 16:34:16
Round 2
Reviewer 2 Report
The sample size calculation was added, and it seems that the points pointed out by me have been corrected.